# The Interplay of Structuring and Controlling Teaching Styles in Physical Education and Its Impact on Students’ Motivation and Engagement

**DOI:** 10.3390/bs14090836

**Published:** 2024-09-18

**Authors:** Javier Coterón, José Fernández-Caballero, Laura Martín-Hoz, Evelia Franco

**Affiliations:** 1Social Sciences Applied to Physical Activity, Sport and Leissure Department, Faculty of Physical Activity and Sport Sciences-INEF, Universidad Politécnica de Madrid, 28040 Madrid, Spain; j.coteron@upm.es (J.C.); jose.fernandez-caballero-ruiz@alumnos.upm.es (J.F.-C.); laura.martin.hoz@alumnos.upm.es (L.M.-H.); 2Communication and Education Department, Faculty of Health Sciences, Universidad Loyola Andalucía, 41704 Seville, Spain

**Keywords:** teaching styles, motivation, circumplex model, basic psychological needs, cluster analysis

## Abstract

Background: Teaching style has a significant influence on students’ learning outcomes. This study focused on identifying teaching profiles in Physical Education characterized by high directiveness, using structure and control behaviors that impact students’ outcomes, basic psychological needs (BPN), and engagement. It was based on the circumplex model and self-determination theory (SDT) and intended to explore how these styles affect students’ motivation and engagement. Methods: A cluster-based methodological design was applied, evaluating teachers through self-reports. Adapted measures of structure and control were used to classify teachers into four distinct profiles within the educational context of Physical Education. Results: The study identified three teaching profiles: ‘high structure–low control’, ‘high structure–high control’, ‘low structure–low control’, and ‘low structure–high control’. The ‘high structure–low control’ profile showed the best results in autonomous and controlled motivation, with greater behavioral engagement among students. In contrast, the ‘high structure–high control’ profile was associated with higher levels of demotivation. Conclusions: Teaching styles of structure and control can combine in various ways among Physical Education teachers, significantly influencing student motivation, satisfaction of basic psychological needs, and engagement. It is recommended that teachers adopt behaviors that support structure without becoming controlling to enhance student learning and participation in classes.

## 1. Introduction

The quality of experiences in Physical Education (PE) class is crucial for fostering interest in physical activity and the desire to spend leisure time in such activities [1]. Numerous studies have highlighted that both the engagement and motivation experienced by adolescents in PE class are associated with various behavioral, cognitive, and affective outcomes in both PE and other physical activity contexts.

### 1.1. Students’ Engagement in PE

Student engagement is a multifaceted concept encompassing behavioral, emotional, and cognitive dimensions [2]. Engaged students actively listen, put forth effort, persist in tasks, respond to questions, and enjoy participating in activities [3,4]. Conversely, disengaged students lack effort and persistence, easily give up, do not listen to the teacher, and experience boredom [5]. Behavioral engagement in the class has been defined as students paying close attention to sources of information, investing effort, and persisting in the face of setbacks. On the other hand, agentic engagement refers to students’ proactive involvement in shaping the instruction they receive. It encompasses actions and expressions aimed at creating a more motivationally supportive learning environment for themselves [6,7,8].

Both types of engagement have received attention from researchers and educators in PE since they have been found to be related to other desirable outcomes, such as students’ learning, skill development, and adherence to physical activity (PA) [9]. Evidence suggests that engagement is influenced by teacher–student interactions [10], and several studies have examined this relationship through the lens of self-determination theory (SDT) [11].

### 1.2. Self-Determination Theory: A Key Perspective to Gain Understanding of the PE Setting

SDT [11] has frequently been used as a valid framework to explore the PE setting and understand the processes through which students’ experiences in this context might impact their perceptions and behaviors [12]. This theoretical approach posits that students can experience different motivational regulators based on their levels of self-determination, identified as autonomous motivation (related to engaging in an activity for inherent pleasure or personal benefit), controlled motivation (when students feel guilt for not performing an action or participate for external rewards), and amotivation (lack of reasons to engage in certain PE activities). According to SDT, students are more likely to experience self-determined forms of motivation when their BPN are fulfilled. SDT identifies three BPN: autonomy (feeling of choice in activities or having a say in class matters), competence (feeling effective or capable of achieving proposed activities), and relatedness (feeling accepted by peers and being able to confide in both students and the teacher). The satisfaction of these needs leads to optimal psychological functioning and the most self-determined forms of motivation [13,14]. Research shows that a need-supportive context best supports students’ motivation through the satisfaction of their psychological needs [15]. Such a context nurtures students’ BPN, with teachers implementing strategies to foster each of the three needs [16,17].

The frustration of BPN refers to the negative feelings experienced when a person perceives that their needs are being actively undermined by someone’s actions in their close social environment [14,18]. Autonomy frustration occurs when students have no choice in activities, cannot progress at their own pace, or feel their interests are disregarded. Competence frustration arises when students receive critical feedback in front of peers, lack individualized challenges, or are grouped by ability. Relatedness frustration occurs when students feel ignored and excluded from their peer group, for instance, during team selection. When the context threatens their BPN, students exhibit more controlled motivation, feeling pressured to participate in activities. Although BPN satisfaction has been more extensively studied, highlighting its importance in promoting motivation [19], evidence also shows that BPN threat or frustration leads to less self-determined motivation, affecting both students’ and teachers’ well-being and their interactions [14,20].

### 1.3. The Circumplex Approach and the Role of Teaching Styles in PE

Recently, utilizing SDT perspective, an innovative approach has been developed to enhance the understanding of teacher–student interactions through the analysis of teaching behaviors [21,22]. As illustrated by Aelterman et al. [21], Figure 1, the model distinguishes four different dimensions based on the vertical axis (level of directiveness) and the horizontal axis (level of needs support provided). Notably, this approach does not consider the satisfaction or frustration of each need independently but rather establishes the horizontal axis as a measure of overall need support.

When teachers exhibit needs-supportive behaviors with low directiveness, they employ an autonomy-supportive teaching style. Conversely, when teachers demonstrate needs-supportive behaviors with high directiveness, they adopt a structured teaching style. On the other hand, teachers who display needs-thwarting behaviors with high directiveness implement a controlling teaching style, while those exhibiting needs-thwarting behaviors with low directiveness display a chaotic teaching style.

Within the autonomy-support dimension, teachers can adopt participative strategies, such as identifying students’ personal interests and offering choices, or attuning strategies by making activities more interesting and enjoyable to nurture students’ interests. Regarding structure, teachers can utilize guiding practices, such as suggesting individual progressions to help students complete tasks, or clarifying strategies by being transparent and clear about lesson expectations. A controlling teaching style is characterized by demanding strategies, like using authoritative language to enforce discipline and changing students’ thoughts, or domineering strategies, inducing feelings of guilt and shame to ensure compliance. In a chaotic style, teachers may employ abandoning strategies, leaving students on their own, or awaiting strategies, where teachers do not plan lessons and create contexts where students must take the initiative.

### 1.4. The Merits of a Person-Centered Approach to Better Understand the Coexistence of Structure and Control

Providing students with clear instructions is considered crucial for effective classroom management [23]. In this regard, PE teachers can employ a wide variety of strategies [12,24]. Based on SDT [25], competence support and control are two styles characterized by high teacher directiveness. However, SDT describes how these two styles are qualitatively different: one supports needs (structure), while the other hinders them (control) [26]. Thus, teachers can guide students in two qualitatively distinct ways, likely used simultaneously as classroom management strategies [27,28,29].

In the last decade, SDT-based research has documented the differentiated effects of teachers’ structure and control styles in PE classes [12]. However, there is limited evidence on the distinctive role of structure from the students’ perspective [12]. Recent research has examined the effects of teachers’ controlling style compared to autonomy-supportive style in PE lessons [12,24,30]. Given that the concept of competence support has emerged more recently in research, little is known about the potential combined effects of structure and control styles in PE classes. Considering that both structure and control refer to highly directive styles (i.e., in which the teacher leads learning interactions), PE teachers often question how to effectively implement a structured style for optimal classroom management without resorting to controlling practices. Thus, new research is needed to address how teaching styles with high directiveness (structure and control), separately or combined, influence student motivation and engagement.

Although most studies indicate that needs-supportive behaviors are common among PE teachers, needs-thwarting behaviors are also present [31,32]. For example, teachers who pressure students to meet their demands may also provide valuable instructions and feedback. Therefore, classifying teachers as exclusively needs-supportive or needs-thwarting may be inaccurate. Previous studies have examined the relationship between these behaviors and students’ motivational outcomes using variable-centered approaches [12]. SDT-based studies have identified that PE teachers can combine various motivating styles in their practice [27,28,29], highlighting four distinct profiles in terms of needs-supportive and needs-thwarting behaviors. Two common profiles are high support and low thwarting, and low support and high thwarting. The other two profiles vary in combinations of support and thwarting. Person-centered approaches, in contrast to variable-centered approaches, assume that the relationships between variables can differ among individuals [33]. Theoretically, these approaches could clarify whether supportive and thwarting behaviors represent opposite extremes of the same continuum or are distinct but correlated dimensions.

### 1.5. The Present Study

So far, no studies have specifically examined the combinations of structure and control teaching styles in PE and how they affect student outcomes. Since excessive competence support can be perceived as control and vice versa, further research is needed on how teachers guide students and whether the style in which instructions are given is important. More evidence is also required on the separate and combined roles of these two styles in relation to students’ experiences and motivation in PE.

The aim of this study is twofold. First, it seeks to establish profiles of PE teachers based on their teaching style, characterized by high directiveness, represented by structure and control styles. Four distinct profiles are expected to be identified: two with opposing levels of structure and control (‘high structure–low control’ and ‘low structure–high control’), and two where both styles are combined at similar levels (‘high structure–high control’ and ‘low structure–low control’).

The second objective is to examine the potential differences in students’ motivation and engagement based on their teachers’ profiles. It is hypothesized that students taught by teachers displaying high structure and low control levels will present the most adaptive outcomes (high autonomous motivation and engagement), while those students taught by teachers exhibiting low levels of structure and high levels of controlling behaviors will display high levels of controlled motivation and amotivation, and lower levels of autonomous motivation and engagement. Given the lack of studies addressing how differently displaying either high or low levels of structure and control may impact students’ outcomes, we are cautious in making assumptions about the different consequences of these two profiles.

## 2. Materials and Methods

### 2.1. Design and Participants

The present study employed a cross-sectional design with a non-probabilistic convenience sample. A total of 56 secondary school PE teachers aged 23–59 years (M = 37.54, SD = 10.55), with an average teaching experience of 10.59 years (SD = 10.22), participated, of whom 28.6% were female. Additionally, 542 secondary school PE students aged between 12 and 17 years (M = 13.48, SD = 1.22) participated, of whom 49.2% were girls. An inclusion criterion for participants was that teachers should have taught the corresponding group for at least one full academic year. Participants were recruited from schools located in the Autonomous Community of Madrid.

### 2.2. Measures

*Teaching styles* as perceived by teachers were assessed through a version of the Situations in School (SIS) questionnaire designed by Aelterman et al. [21] adapted for the Spanish context in PE (SIS–PE) [26,34]. The scale consists of 12 vignettes of common situations in class. For each of the 12 situations, there are four distinct responses (i.e., 48 items), with each representing an autonomy-supporting, structuring, controlling, and chaotic style. The SIS–PE questionnaire includes four participative and eight attuning items for a total of 12 autonomy-support items; seven guiding and five clarifying items together constitute 12 structure items; seven demanding and five domineering items form 12 control items; and finally, eight abandoning and four awaiting items make up a total of 12 chaos items. Reactions to each situation were provided on a seven-point Likert scale ranging from 1 (“does not describe me at all”) to 7 (“describes me extremely well”). This tool had been previously validated with a sample of 250 PE teachers, and the fit indices for the eight-dimension confirmatory factor analysis were adequate (χ^2^[224] = 497.44; *p* < 0.001; χ^2^/df = 2.22; CFI = 0.91; IFI = 0.90; RMSEA = 0.06; SRMR = 0.05), the Cronbach alphas ranging from 0.77 to 0.94.

*Students’ basic psychological need satisfaction and frustration* were measured with the Spanish version for the PE context of the Basic Psychological Need Satisfaction and Frustration Scale in Physical Education [35,36]. The items were introduced by the stem “In my PE class…”. This instrument includes 24 items (four items per dimension) and captures autonomy satisfaction (e.g., “I feel my choices express who I really am”) and frustration (e.g., “I feel forced to do many things I wouldn’t choose to do”), competence satisfaction (e.g., “I feel competent to achieve my goals”) and frustration (e.g., “I feel insecure about my abilities”), and relatedness satisfaction (e.g., “I feel close and connected with other people who are important to me”) and frustration (e.g., “I feel the relationships I have are just superficial”). The items were rated on a five-point Likert scale from 1 (strongly disagree) to 5 (strongly agree). In this study, CFA for six factors showed an adequate fit: χ^2^(237) = 795.07, *p* < 0.001, χ^2^/df = 3.36; CFI = 0.96; TLI = 0.95; SRMR = 0.04; RMSEA = 0.04 (90%CI = 0.03–0.04). Adequate internal consistency values were also obtained in Cronbach’s alpha and omega ranging from 0.81 to 0.86 in both cases.

*Students’ motivation* was measured using the Spanish version for the PE context of the Perceived Locus of Causality Scale (PLOC) [37,38,39]. The items were introduced by the stem “I take part in PE classes…”. This scale includes 24 items reflecting six different types of motivation: intrinsic motivation (e.g., “Because I enjoy learning new skills”; α = 0.85; ω = 0.85), integrated regulation (e.g., “Because it agrees with my way of life”; α = 0.88; ω = 0.89), identified regulation (e.g., “Because it is important for me to do well in PE”; α = 0.82; ω = 0.82), introjected regulation (e.g., “Because I would feel bad with myself if I did not do it”; α = 0.77; ω = 0.77), external regulation (e.g., “Because that is what I am supposed to do”; α = 0.71; ω = 0.70), and amotivation (e.g., “But I do not understand why we should have PE.”; α = 0.84; ω = 0.84). Students responded to the different reasons using a five-point Likert scale from 1 (strongly disagree) to 5 (strongly agree). Following SDT tenets, autonomous motivation was calculated as the means of intrinsic motivation, integrated regulation, and identified regulation; and controlled motivation was calculated as the means of introjected and external regulation. In this study, the CFA for three factors (autonomous motivation, controlled motivation, and amotivation) showed an adequate fit: χ^2^(237) = 1189.17, *p* < 0.001, χ^2^/df = 5.02; CFI = 0.93; TLI = 0.92; SRMR = 0.05; RMSEA = 0.05 (90%CI = 0.05–0.06). Adequate internal consistency values were also obtained in Cronbach’s alpha and omega ranging from 0.71 to 0.88, and 0.70 to 0.89, respectively.

*Students’ behavioral engagement* was measured with the Spanish version [17] of the scale adapted from Shen, McCaughtry, Martin, Fahlman, and Garn [40]. The stem used in the questionnaire was “in PE classes”, and it was followed by 5 items addressing students’ perceptions of their effort, attention, and persistence in PE classes (e.g., “I work as hard as I can”). Responses were given on a five-point scale ranging from 1 (strongly disagree) to 5 (strongly agree). Cronbach’s alpha was 0.87.

*Students’ agentic engagement* was measured with the Spanish version [41] of the scale developed by Reeve [7]. This instrument is composed of 5 items that measure the construct of agentic engagement as a single factor (e.g., “During class, I share my preferences and opinions”). Responses were given on a five-point scale ranging from 1 (strongly disagree) to 5 (strongly agree). Cronbach’s alpha was 0.72.

### 2.3. Procedure

The study received the approval of a Spanish University Ethics Committee (code 2022/46) and was conducted in accordance with the ethical guidelines established by the American Psychological Association [42], guaranteeing informed consent, confidentiality, and anonymity of participants’ responses. After approval was received, the heads of department and PE teachers from the different participating schools were contacted and informed about the study’s objectives and their collaboration was requested. Having identified the centers willing to participate in the study, teachers, students, and students’ parents or legal guardians were also informed about the study. Once informed consent was obtained from the participating teachers and the students’ families, data were collected. A research assistant agreed with participant teachers on a day that was convenient for them to attend his or her school. Fifteen minutes before the end of the session, students were asked to honestly answer a printed questionnaire after the research assistant explained to them the purpose of the study and highlighted that there were no right or wrong answers. Teachers were personally addressed through being asked to answer an online questionnaire, so that they could take time to reflect on their answers. Both students and PE teachers were provided with a random key composed of letters and/or numbers which was included in the questionnaire to allow for subsequent data association (e.g., GOL19). It was the research assistant who assigned these codes. In order to guarantee the anonymity of the responses, data were treated by other members of the research team who did not have access to the code assignation made by the aforementioned research assistant.

### 2.4. Data Analysis

Descriptive statistics (mean and standard deviation) and correlations among all the study variables were calculated. Internal consistency (i.e., reliability) was examined via Cronbach’s alpha and coefficient omega (α and ω > 0.70; [43]). Furthermore, CFAs were performed to test the factorial structure of the instruments used to gather the data. To assess the models’ fit, the following indices were used: χ^2^, df, CFI, TLI, RMSEA, and SRMR. Scores greater than 0.90 for incremental indices such as CFI and TLI are acceptable [44]. A model is considered to have a good fit if the RMSEA and SRMR values are less than 0.08 and 0.06, respectively [44]. Subsequently, quartiles were calculated on the structure and control scores of the 56 teachers, and groups were formed of teachers whose scores on the two variables were in the extreme quartiles (teachers who scored on either variable in the 2nd and 3rd quartiles were discarded). This analysis resulted in three groups of teachers, since no teacher scored in the 4th quartile of structure and in the 1st quartile of control. The standardized scores for structure and control and chaos styles were calculated to better characterize and label the three resulting profiles. To analyze the differences in students’ outcomes according to their teachers’ profiles, the students taught by teachers identified in the previously described profiles were selected, and this sample was split into three groups in consonance with the cluster in which their teachers had been placed. A multivariate analysis of variance (MANOVA, Wilks’ lambda test) with the cluster solution as independent variable were used to investigate the differences in students’ motivation, need satisfaction, need frustration, behavioral engagement, and agentic engagement. Post hoc tests by means of Bonferroni method were inspected if significant differences were found. Effect sizes were considered small, moderate, or large, when partial eta-squared values were above 0.01, 0.06, and 0.14, respectively.

## 3. Results

### 3.1. Preliminary Analyses and Descriptive Statistics

After calculating the quartiles for the structure and control scores of the 56 teachers, and forming groups based on their standardized scores, a group of 19 teachers with a “structure–control” profile was identified, constituting the final teacher sample for analysis (Table 1). Consequently, students taught by these 19 teachers were selected from the general sample as the final student sample. As a result, the analyzed sample comprised 540 students, distributed as shown in Table 1.

Table 2 presents the descriptive statistics and bivariate correlations of the variables used for both teachers and students in the study. Overall, the correlations indicated significant and strong relationships among all the measured study variables.

### 3.2. Teaching Profiles According to Teachers’ Directiveness

The quartile-based analysis of structure and control teaching behaviors revealed the existence of three distinct profiles of highly directive teachers. Based on their standardized relative scores (Table 3 and Figure 1), the following labels were assigned to the three groups: (a) a ‘low structure–low control’ group (*n* = 6), characterized by low levels of both control and structure teaching style behaviors; (b) a ‘high structure–high control’ group (*n* = 11), characterized by high levels in the use of both structure and control teaching style behaviors; and (c) a ‘high structure–low control’ group (*n* = 2), characterized by high levels in structure teaching style behaviors and low levels in control teaching style behaviors. Figure 1 illustrates the graphical results of the three groups, with different teachers associated by means of Z-scores. Significant multivariate differences were found between clusters (F_14,20_ = 5.48, *p* < 0.001, η^2^ partial = 0.793), confirming the established labelling. Univariate differences and between-cluster contrasts are presented in Table 3.

After conducting the ANOVA, significant differences were found among the three groups of teachers across all variables related to teaching styles (Table 3). The differences between the ‘low structure–low control’ group and the ‘high structure–high control’ group were significant across all teaching style variables, aligning with the literature, as these two groups represent the extremes within these styles. Additionally, the ‘high structure–low control’ style showed significant differences compared to the other two profiles in variables corresponding to the opposite extremes of each style.

### 3.3. Differences in Students’ Outcomes

As depicted in Table 4, students whose teachers exhibit a ‘high structure–low control’ profile achieve higher scores in autonomous motivation (M = 4.06) and controlled motivation (M = 3.58), while displaying lower levels of amotivation (M = 1.76). These differences are statistically significant compared to the other two profiles (M_autonomous motivation–high structure, high control_ = 3.70; M_controlled motivation–low structure, low control_ = 3.27; M_controlled motivation–high structure, high control_ = 3.31; M_amotivation–low structure, low control_ = 2.05; M_amotivation–high structure, high control_ = 2.18), except for autonomous motivation in students with ‘low structure–low control’ teachers (M = 3.74).

Regarding overall BPN satisfaction, students taught by teachers in the ‘high structure–low control’ profile reported higher scores (M = 3.81) than those in the ‘high structure–high control (M = 3.48) and in the ‘low structure–low control’ (M = 3.40) profiles. More specifically, profiles with high structure obtain better results in relatedness satisfaction (M_high structure–high control_ = 3.63; M_high structure–low control_ = 3.82) than students taught by teachers in the ‘low structure–low control’ group (M = 3.40). Also, the ‘high structure–low control’ profile obtained the highest levels of autonomy satisfaction among students (M = 3.67).

On the other hand, as for BPN frustration, patterns are contrary to what is observed in BPN satisfaction, and students taught by teachers in the ‘high structure–low control’ group displayed lower scores (M = 2.04) than those in the ‘high structure–high control’ (M = 2.44) and in the ‘low structure–low control’ (M = 2.30) groups. Specific findings show that the ‘high structure–low control’ profile displayed lower levels than the ‘high structure–low control’ profile in both autonomy (M_high structure–low control_ = 2.14 vs. M_high structure–high control_ = 2.55) and competence frustration (M_high structure–low control_ = 2.00 vs. M_high structure–high control_ = 2.44) among students.

Finally, as for student engagement, scores were higher in students taught by teachers in the ‘high structure–low control’ profile (M = 3.98) when compared with those in the ‘high structure–high control’ (M = 3.58) and in the ‘low structure–low control’ (M = 3.74) profiles. It was specifically in the behavioral engagement dimension that scores were higher among students whose teachers had a ‘high structure–low control’ style (M = 4.33) compared to students taught by both teachers in the ‘low structure–low control’ profile (M = 3.96) and in the ‘high structure–high control’ group (M = 3.87).

## 4. Discussion

### 4.1. Teaching Profiles of Highly Directive Teachers

The primary objective of this study was to establish profiles of PE teachers based on their teaching styles, particularly those characterized by high directiveness, using their scores on behaviors associated with structure and control styles. Findings revealed the existence of three different teachers’ profiles labelled as ‘low structure–low control’, ‘high structure–high control’ and ‘high structure–low control’. Overall, these results align with other research that has analyzed motivational profiles based on the support and threat to BPN, suggesting that PE teachers can combine various motivational styles in their practice [27,28], although there were four profiles identified when autonomy support and control style, or BPN support and threat, were considered to establish the groups.

Recently, in close connection with the present study, García-González et al. [45] studied the combination of structure-related teaching behaviors with controlling behaviors, differentiating between internal and external control. They identified four profiles (‘high structure–low control’, ‘moderate structure–moderate control’, ‘moderate structure–high control’, and ‘low structure–very high control’) based on student perceptions. While the first two of these profiles concur with two of the groups established in the present study, interesting differences emerged between the solution reached by García-González et al. [45] and the solution in the present study. More specifically, the absence in the present study of a profile displaying a low level of structure and high level of control is noteworthy. This result suggests that while a teacher might exhibit structured behaviors without control, it is unlikely for a teacher self-perceived as controlling not to also be self-perceived as structured. This finding leads us to think that when teachers report their own behaviors, a ‘low structure–high control’ profile might not be identified, possibly, because teachers who use controlling behaviors often do so with the intention of guiding or clarifying. However, when perceptions come from students, as in García-González et al. [45], students can identify controlling behaviors in teachers with a low structure profile. This discrepancy may be due to the fact that some teacher behaviors intended to guide or clarify may not be perceived as such by students and are instead seen merely as controlling. It could be the case for behaviors oriented to maintain discipline in class. Student misbehavior has consistently been regarded as one of the biggest concerns for PE teachers [46,47], and disciplined behaviors in class seem to be crucial to achieve a successful learning process [48,49] (Gutiérrez and López, 2012; Wang et al., 2010). This fact might explain the frequent display of structure and control strategies by PE teachers [50]. It could happen thus that when teachers emphasize the relevance of maintaining order in class, and/or provide continuous instructions so that all students behave according to their expectations, they do so in the belief that it will positively impact their students’ learning, while it is perceived by students as a form of controlling behavior.

Furthermore, it may be that certain teacher behaviors, even if infrequent, have a significant impact on students’ experiences. Survey and observational tools designed to analyze teaching behaviors from a circumplex approach tend to quantify the number/level of appearance for each fixed behavior, and data are analyzed assuming that the more frequent a behavior is, the more impact it might have on students’ experiences [26,51,52]. However, there is evidence that when teachers adopt their own perspective and can use their own words to describe how learning situations are perceived in the PE context, a richer and more comprehensive picture can be taken by focusing on those events which might be critically relevant to shaping students’ experiences [53]. Further research in this line could shed some light on the reasons underlying why teachers rely on highly directive behaviors.

### 4.2. Differences in Student Outcomes and Engagement According to Teacher Profile

The second objective of the study was to examine potential differences in student outcomes based on their teachers’ profiles and how these relate to student motivation and engagement. Only one previous study has analyzed the influence of PE teacher profiles characterized by levels of directiveness [45]. They found that students who perceived their teachers as more competence-supportive and rarely using controlling practices reported the highest levels of need satisfaction and autonomous motivation, as well as the lowest levels of need frustration, controlled motivation, and amotivation. Furthermore, they reported that high levels of directiveness are not necessarily positive if the guidance provided predominantly frustrates students’ needs. However, this study focused on students’ perceptions and nested the sample by classrooms without differentiating groups that shared the same teacher. Our study offers a novel approach by associating the teacher’s profile with their specific group of students, strengthening the cause–effect relationship between teacher profile and student outcomes. Additionally, we identified teacher profiles through their own perceptions, providing a different perspective.

Our findings are consistent with the assumptions of SDT [14] and previous studies in PE [12,54], showing that profiles with high structure are associated with greater satisfaction of BPN and lower threat to these needs. An especially interesting aspect of our results is the absence of significant differences in competence satisfaction among the profiles, but significant differences in competence frustration. This suggests that implementing controlling behaviors does not provide additional benefits in terms of competence satisfaction compared to the ‘low structure–low control’ profile. However, these controlling behaviors can increase competence frustration. A structured yet non-controlling teaching style is characterized by providing clear guidelines and consistent support without resorting to coercion or strict rule imposition. For instance, a teacher may set clear expectations and offer specific and constructive feedback while allowing students some autonomy in decision-making on how to achieve those objectives. This approach fosters a learning environment where students feel supported but not pressured, optimizing their motivation and engagement. In other words, while a highly directive style may support structure, it is not beneficial if it also includes controlling behaviors, as it does not enhance competence satisfaction and, in fact, can increase frustration. Consistent with existing literature on motivation [12,13], the ‘high structure–low control’ profile is associated with higher levels of autonomous and controlled motivation, as well as lower levels of amotivation. In the other two profiles, the levels of motivation (both autonomous and controlled) and amotivation were similar.

Regarding student engagement, a crucial aspect of student involvement in proposed tasks, we observed that the ‘high structure–low control’ profile yields the best results in behavioral engagement, helping to create a more conducive classroom climate for learning. This is consistent with Coterón et al. [55] and Xiang et al. [56], who linked higher student motivation with greater engagement levels. Additionally, although no significant difference was found, this profile also achieves higher levels of agentic engagement in absolute terms, highlighting that controlling behaviors can reduce this type of engagement. In line with the findings of Reeve and Shin [57], teacher autonomy support promotes students’ motivational satisfaction, which in turn fosters greater agentic engagement. In contrast, controlling styles can generate motivational frustration, thereby diminishing students’ capacity to engage actively and agentically in learning. Thus, promoting a teaching style that supports autonomy without resorting to control may be key to maximizing both behavioral and agentic engagement in students.

The novelty of the design and approach of this research makes it challenging to compare the results in their entirety. Nevertheless, when compared with other studies that have addressed different aspects individually or from other perspectives, we find that the best results in autonomous and controlled motivation, as well as the highest levels of BPN satisfaction and the lowest threat to these needs, correspond to students whose teachers are characterized by greater structure [27,28,29,45]. This indicates that teachers who guide students in task completion and clarify learning expectations and objectives achieve more adaptive outcomes. On the other hand, the ‘high structure–high control’ profile is associated with the highest levels of amotivation and the lowest levels of autonomous and controlled motivation, consistent with previous studies [31]. However, this profile is also related to greater satisfaction and lower frustration of the need for relatedness, in line with García-González et al. [45].

### 4.3. Practical Implications

In practical terms, the present study illuminates several educational concerns and deepens the understanding of principles established by SDT and the circumplex model, expanding their knowledge and application. Combined with other research, such as that of García-Cazorla et al. [58], which observed that controlling behaviors of PE teachers are related to the frustration of basic psychological needs (BPN) and student demotivation, this study advances the understanding of the processes influencing the educational environment and teacher–student relationships to improve teaching outcomes.

Firstly, it demonstrates that teachers’ profiles do not strictly adhere to a single teaching style proposed by the circumplex model. This opens avenues for investigating the potential relationships between the behaviors and profiles proposed in this model and the combinations that can occur in PE teachers, considering that these behaviors may fluctuate throughout the class or due to other conditions.

Based on the results obtained, PE teachers are recommended to use behaviors associated with the structure teaching style, as this approach yields better outcomes for students in terms of motivation, satisfaction, BPN support, and engagement. However, it is also noted that control behaviors may have certain benefits when combined with structure behaviors, although their use should be significantly limited.

To create a learning environment supported by structure behaviors and avoid control behaviors, previous research recommends that PE teachers implement student-centered pedagogical models such as Sport Education, Teaching Games for Understanding, or teaching styles like guided discovery, while avoiding the use of direct instruction models [59,60,61]. Therefore, the results provide an evidence base for planning and developing various programs based on SDT to train and empower PE teachers to improve the behaviors employed during classes, thereby enhancing student outcomes.

### 4.4. Limitations and Future Research Directions

The present study has several limitations that should be considered when interpreting the findings. Firstly, the sample size of teachers is small after grouping them into the corresponding clusters. A larger sample could provide more consistent results and reinforce the finding that a low-structure and high-control teaching style is unlikely, thus increasing the representativeness and generalizability of the findings. Additionally, the cross-sectional design of the study limits the ability to draw conclusions about causal effects, as the direction of the observed relationships cannot be determined. Future studies adopting an experimental design could further investigate how teaching profiles influence outcomes, the satisfaction of or threat to BPN, and student engagement during PE classes, providing a more robust understanding of these dynamics.

Although the use of measures including self-reports from both teachers and students overcame some limitations noted in previous studies that relied exclusively on teacher self-reports (e.g., Van den Berghe, Soenens, et al. [62]), some concerns remain. Additional objective observations, particularly regarding behaviors related to teaching practices and student outcomes, could eliminate potential perceptual biases and provide a more accurate and holistic evaluation of teaching styles. This would make the differences between teaching profiles even more solid and reliable, contributing to a better understanding of effective practices in PE. Moreover, incorporating multiple evaluation methods could help triangulate the data, strengthening the validity of the study’s conclusions.

The results of this study open a line of inquiry into analyzing possible combinations of teaching styles and behaviors among PE teachers, aiming to understand which are the most common, their characteristics, and the outcomes they generate for students. Furthermore, it is worth investigating how teaching styles and behaviors may change during a class or based on other factors that might influence a shift, rather than considering them as fixed profiles that always act in the same manner. Given that the sample in this study can be considered small, replicating the study with a larger sample and/or considering other factors such as geographical location or type of educational institution would be valuable to deepen the results obtained and enhance their representativeness. Lastly, a research design that uses various tools for data collection, in addition to self-reports, by combining these with external objective observations and/or student evaluations of teaching styles, could expand the understanding of teaching practices and their outcomes. Incorporating additional evaluation methods would strengthen the conclusions drawn from the results and provide a more holistic and accurate view of teaching practices in PE.

## 5. Conclusions

The results of the present study suggest that various teaching styles identified within the circumplex model can combine to create different teacher profiles, which in turn influence student outcomes in terms of motivation, satisfaction of and threat to BPN, and engagement. Main findings and practical implications are summarized in Figure 2.

In particular, the teaching styles in the Circumplex model characterized by high directiveness, structure, and control can merge in ways that result in distinct profiles. The profile characterized by high structure and low control stands out as the most effective, achieving the highest levels of autonomous and controlled motivation, reducing amotivation, satisfying the needs for autonomy and relatedness, minimizing the frustration of autonomy and competence needs, and securing greater behavioral engagement.

This study enhances the understanding of how different teaching styles and behaviors can combine to positively influence students, providing a foundation for future research and practical applications in the field of Physical Education.

## Figures and Tables

**Figure 1 behavsci-14-00836-f001:**
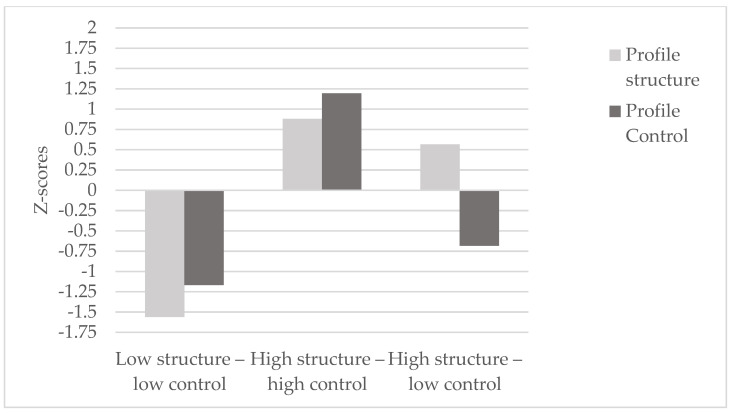
Three-group solution based on Z-scores for structure and control profiles among secondary PE teachers. The absolute values are provided in Table 3.

**Figure 2 behavsci-14-00836-f002:**
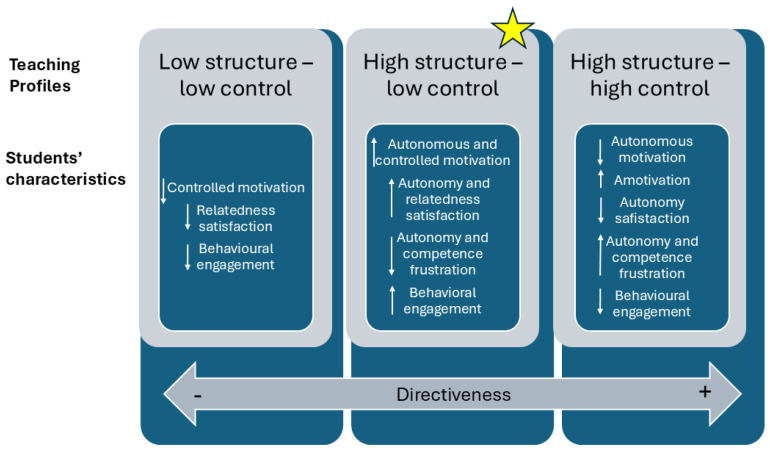
Findings summary. Star indicates the most adaptive teaching profile identified in the study. Up arrows reflect relatively higher scores in the corresponding outcome and down arrows reflect relatively lower scores in the corresponding outcome.

**Table 1 behavsci-14-00836-t001:** Frequency of teachers corresponding to each profile and frequency of associated students.

Teaching Profile	N (Teachers)	N (Students)
Low structure–low control	6	162
High structure–high control	11	326
High structure–low control	2	52

**Table 2 behavsci-14-00836-t002:** Descriptive statistics and correlations among study variables for students and teachers.

Students’ Variables	1	2	3	4	5	6	7	8	9	10	11
1 Autonomous motivation	-										
2 Controlled motivation	0.483 **	-									
3 Amotivation	−0.234 **	0.142 **	-								
4 Autonomy satisfaction 5 Competence satisfaction	0.647 **	0.418 **	−0.099 *	-							
	0.528 **	0.238 **	−0.136 **	0.567 **	-						
6 Relatedness satisfaction	0.540 **	0.356 **	−0.154 **	0.663 **	0.473 **	-					
7 Autonomy frustration	−0.320 **	0.144 **	0.469 **	−0.269 **	−0.269 **	−0.246 **	-				
8 Competence frustration	−0.166 **	0.250 **	0.350 **	−0.131 **	−0.313 **	−0.148 **	0.598 **	-			
9 Relatedness frustration	−0.038 **	0.184 **	0.308 **	−0.015	−0.165 **	−0.156 **	0.434 **	0.570 **	-		
10 Behavioral engagement	0.383 **	0.332 **	−0.124 **	0.300 **	0.243 **	0.336 **	−0.124 **	−0.023	0.005	-	
11 Agentic engagement	0.474 **	0.321 **	−0.067	0.510 **	0.456 **	0.433 **	−0.172 **	−0.152 **	−0.033	0.476 **	-
M	3.75	3.33	2.11	3.19	3.70	3.58	2.50	2.36	2.23	3.94	3.40
*SD*	0.85	0.66	1.07	0.94	0.96	0.98	1.07	1.02	1.02	0.74	0.95
**Teachers’ variables**	**1**	**2**	**3**	**4**	**5**	**6**	**7**	**8**	**9**		
1 Profile structure	-										
2 Guiding	0.882 **	-									
3 Clarifying	0.879 **	0.550 **	-								
4 Profile control	0.418 **	0.207	0.531 **	-							
5 Demanding	0.464 **	0.301 *	0.518 **	0.921 **	-						
6 Domineering	0.327 *	0.101	0.476 **	0.944 **	0.741 **	-					
7 Autonomy support	0.318 *	0.334 *	0.225	−0.014	0.076	−0.089	-				
8 Competence support	0.495 **	0.512 **	0.358 **	−0.014	0.122	−0.128	0.795 **	-			
9 Relatedness support	−0.201	−0.231	−0.123	0.052	−0.019	0.106	−0.154	−0.174	-		
M	5.42	5.84	5.01	3.36	3.82	2.90	6.28	6.21	1.37		
*SD*	0.67	0.76	0.75	1.00	0.99	1.17	0.92	0.91	0.81		

Note: * = *p* < 0.05; ** = *p* < 0.01.

**Table 3 behavsci-14-00836-t003:** Three-group solution: mean scores of the teaching style clusters and F-values for teacher variables.

Teachers	Low Structure–Low Control (*n* = 6)	High Structure–High Control (*n* = 11)	High Structure–Low Control (*n* = 2)	*F* (df)	η^2^ Partial
*Structure*	4.41 (0.90) ^a^	6.04 (0.24) ^b^	5.83 (0.00) ^b^	18,18 (2) ***	0.694
Guiding	4.93 (1.04) ^a^	6.34 (0.33) ^b^	5.86 (0.00)	9.48 (2) **	0.542
Clarifying	3.90 (0.79) ^a^	5.75 (0.53) ^b^	5.80 (0.00) ^b^	19.11 (2) ***	0.705
*Control*	2.14 (0.32) ^a^	4.61 (0.60) ^b^	2.65 (0.11) ^a^	50.22 (2) ***	0.863
Demanding	2.33 (0.48) ^a^	4.89 (0.61) ^b^	3.00 (0.20) ^a^	43.49 (2) ***	0.845
Domineering	1.97 (0.48) ^a^	4.33 (0.85) ^b^	2.30 (0.42) ^a^	22.62 (2) ***	0.739

Note: standard errors are reported in parentheses; ** *p* < 0.01; *** *p* < 0.001. The mean of one group differs significantly from the mean of another group if they have a different superscript letter. The lack of a superscript indicates a lack of difference from any other group. For example, in terms of between-group differences in guiding, cluster 1 (a) is significantly different from cluster 2 (b) because they have different superscript letters. However, cluster 3 is not different from either of the other two groups as it has no superscript letter.

**Table 4 behavsci-14-00836-t004:** Four-cluster solution: mean scores of the teaching style clusters and F-values for student variables.

Students	Low Structure–Low Control (*n* = 162)	High Structure–High Control (*n* = 326)	High Structure–Low Control (*n* = 52)	*F* (df)	η^2^ Partial
Autonomous motivation	3.74 (0.91)	3.70 (0.83) ^b^	4.06 (0.78) ^a^	3.98 (2) *	0.015
Controlled motivation	3.27 (0.66) ^b^	3.31 (0.64) ^b^	3.58 (0.72) ^a^	4.54 (2) *	0.017
Amotivation	2.05 (1.12)	2.18 (1.05) ^a^	1.76 (0.93) ^b^	3.86 (2) *	0.014
BPN satisfaction	3.40 (0.86) ^b^	3.48 (0.79) ^b^	3.81 (0.70) ^a^	5.19 (2) ***	0.019
Autonomy satisfaction	3.06 (0.98) ^b^	3.17 (0.92) ^b^	3.67 (0.85) ^a^	8.62 (2) **	0.031
Competence satisfaction	3.74 (0.99)	3.63 (0.95)	3.94 (0.85)	2.51 (2)	0.009
Relatedness satisfaction	3.40 (1.03) ^b^	3.63 (0.96) ^a^	3.82 (0.94) ^a^	4.80 (2) **	0.018
BPN frustration	2.30 (0.90) ^a^	2.44 (0.83) ^a^	2.04 (0.86) ^b^	5.43 (2) **	0.020
Autonomy frustration	2.49 (1.12)	2.55 (1.04) ^a^	2.14 (1.09) ^b^	3.28 (2) *	0.012
Competence frustration	2.29 (1.06)	2.44 (0.99) ^a^	2.00 (0.96) ^b^	4.84 (2) **	0.018
Relatedness frustration	2.11 (1.08)	2.33 (1.02)	1.99 (0.90)	4.13 (2) *	0.015
Engagement	3.74 (0.69) ^b^	3.58 (0.73) ^b^	3.98 (0.73) ^a^	7.79 (2) **	0.028
Behavioral engagement	3.96 (0.67) ^b^	3.87 (0.77) ^b^	4.33 (0.63) ^a^	9.16 (2) ***	0.033
Agentic engagement	3.51 (0.96)	3.30 (0.92)	3.62 (1.08)	4.41 (2) *	0.016

Note: standard errors are reported in parentheses; * *p* < 0.05; ** *p* < 0.01; *** *p* < 0.001. The mean of one group differs significantly from the mean of another group if they have different superscript letters. The lack of a superscript indicates a lack of difference from any other group. For example, in terms of between-group differences in autonomous motivation, cluster 2 (a) is significantly different from cluster 3 (b) because they have different superscript letters. However, cluster 1 is not different from either of the other two groups as it has no superscript letter.

## Data Availability

The data presented in this study are available on request from the corresponding author due to ethical reasons (while participants agreed to participate in the study, they did not give explicit and written consent for their data to be shared publicly).

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
