# Peer review of "The Interplay of Structuring and Controlling Teaching Styles in Physical Education and Its Impact on Students’ Motivation and Engagement"

_behavsci, 2024, doi:10.3390/bs14090836_

Round 1
Reviewer 1 Report
Comments and Suggestions for Authors
Comments
1. In Abstract, you mentioned four profiles. However, the manuscript had only three. This is not consistent.
2. In Abstract, “It is recommended that teachers adopt behaviours to enhance student learning and participation in classes.” This sentence is awkward. What do you mean by “adopt behaviors”? What kind of behaviors?
3. In Methods 2.2, “Teaching styles as perceived by teachers were assessed through adapted version for Spanish context in PE ….” Because this study was not anonymous, teachers may hesitate to be totally honest in their self-reports. Low structure is likely perceived as not doing the job. In other words, teachers may tend to provide socially-desired responses. How did you deal with this issue?
4. In 2.3, “anonymity” is mentioned. Because this study needs to match a teacher and his or her class, authors need to explain how “anonymity” can be achieved.
5. In 2.3, “A research assistant agreed with participant teachers on a day that was convenient for them to attend his or her school.” Here some key information is missing. How long had the teacher taught his or her class? This information is crucial. Otherwise, how can we tell students’ motivation, satisfaction and engagement are influenced by this teacher?
6. In 3.2, “Figure X illustrates the graphical results of the three groups….” Check the whole paper to avoid this type of typo. There is no Figure X.
7. In 3.2, “Significant multivariate differences were found between clusters (F (14, 20) = 5.48, p < .001, η2p = .793), confirming the established labelling.” What do 14 and 20 mean in F? What does η2p mean?
8. In Table 3, “Four-cluster solution” was mentioned. But there were only three groups. Should it be three or four-cluster solution?
9. In your sub-title, for example, 3.2 Teaching profiles of highly directive teachers, why do you put “highly” in the sub-title? It is confusing because not all groups were highly directive.
10. What is the purpose for CFA? If you have done CFA, do you think the discriminant validity should be provided? Why not?
11. What is the purpose for CFA? Why do you not conduct a CFA including motivation, satisfaction and engagement all together?
12. For CFA, did you do any modification to improve the model fit? If yes, this is important to let readers know. The reason why this is raised is because your CFA includes positive factor (e.g. motivation) and negative factor (e.g. amotivation). Based on statistical experiences, if a model has both positive and negative factors, the model would have a high chance not to have a good validity. This means CFA may not be so good in model fit.
13. In 3.3, when you try to explain Table 4, you need to tell readers which numbers you are referring to. Otherwise, readers may just guess, but are not sure whether their guess is correct or not.
14. In Table 4, you have 14 factors to discuss for these three groups. This could be too complicated and lose the points. Have you thought of a way to simplify them? For example, is it possible to combine these 14 factors to become, for example, just 3-4 factors? Then, it might be easier to catch your points.
15. I think authors may pay attention not to extend too much in the discussion. For example, Teaching styles were measured only by teachers’ perspective. They are not measured by students or by peers. So if the discussion is beyond this, it is too much and out of the research scope.
16. Let me give you an example to explain what I mean. In Discussion 4.1, “Furthermore, it may be that certain teacher behaviors, even if infrequent, have a significant impact on students' experiences, leading them to identify their teacher as controlling without recognizing support for their competence needs.” This paper did not ask students to identify whether their teachers is controlling or not.
17. Is it possible to provide a table, which summarize the results and discussions or practical implications based on the three different types of teaching styles? This way will help readers to catch your points. In the current version, it is hard to catch the points by long contexts.
Author Response
Dear Reviewer,
Thank you very much for your valuable comments on our work. Please, find attached a document containing our response.

Reviewer 2 Report
Comments and Suggestions for Authors
I wish to begin this brief review by congratulating the authors on their interest, the breadth of approach with which they have presented their work and the precision of their considerations.
Interest in the study of a professional context that demands studies that face learning situations and how physical education teachers can help improve the experience of their students.
The extent of the approach as they have moved away from the usual "labels" which attribute to teachers certain patterns of behaviour based on the label previously assigned. In addition, by including a wide sample of both students and teachers have allowed their results, discussion and conclusions to be of the greatest relevance.
Precision in the sense that they have defined a study design and methodology appropriate to achieving the objectives clearly defined in their proposal.
In conclusion, it is a high quality work. Congratulations to the authors.
Author Response
Dear reviewer,
Please, find attached a document containing our response to your comments.
